# Doppler Ultrasonography of the Fetal Tibial Artery in High-Risk Pregnancy and Its Value in Predicting and Monitoring Fetal Hypoxia in IUGR Fetuses

**DOI:** 10.3390/medicina57101036

**Published:** 2021-09-29

**Authors:** Kristina Norvilaitė, Diana Ramašauskaitė, Daiva Bartkevičienė, Bronius Žaliūnas, Juozas Kurmanavičius

**Affiliations:** 1Center of Obstetrics and Gynecology, Institute of Clinical Medicine, Faculty of Medicine, Vilnius University, 03101 Vilnius, Lithuania; diana.ramasauskaite@mf.vu.lt (D.R.); daiva.bartkeviciene@mf.vu.lt (D.B.); bronius.zaliunas@vgn.lt (B.Ž.); 2Department of Obstetrics, Zurich University Hospital, 8091 Zurich, Switzerland; juozas.kurmanavicius@usz.ch

**Keywords:** IUGR, doppler examination, fetal tibial artery, intrauterine hypoxia, high risk pregnancy

## Abstract

*Background and Objectives*: Intrauterine growth restriction (IUGR) is the term used to describe a fetus whose estimated weight is less than the 10th percentile of its age growth curve. IUGR is the second most common cause of perinatal death. In many cases there is a deficiency in the standardization of optimal management, prenatal follow-up and timing of delivery. Doppler examination is the most sensitive test that can assess the condition of the fetus and indicate fetal intrauterine hypoxia. Numerous studies of the fetal intrauterine state focus on the umbilical artery and the fetal cerebral blood vessels, while the peripheral arteries have so far received insufficient attention. *Materials and Methods*: We present a case of an IUGR fetus monitored with a non-stress test (NST) and a Doppler examination of the fetal arteries (tibial, umbilical, middle cerebral and uterine) and the ductus venosus. In this case the first early sign of fetal hypoxia was revealed by blood flow changes in the tibial artery. *Results*: We hypothesize that peripheral vascular changes (in the tibial artery) may more accurately reflect the onset of deterioration in the condition of the IUGR fetus, such that peripheral blood flow monitoring ought to be employed along with other techniques already in use. *Conclusion*: This paper describes the clinical presentation of an early detection of late IUGR hypoxia and claims that blood flow changes in the tibial artery signal the worsening of the fetus’s condition.

## 1. Introduction

Intrauterine growth restriction (IUGR) is the term used to describe a fetus whose estimated weight is less than the 10th percentile of its age growth curve [1,2,3]. The number of intrauterine deaths is high, not only in early but also in late IUGR [2,3,4]. Therefore, consistent and continuous monitoring of fetal development is crucial so that an early diagnosis can be made. Following the guidelines, surveillance of the middle cerebral artery, umbilical artery Doppler and the cerebroplacental ratio with cardiotocography CTG/NST in IUGR is performed [5,6].

The incidence of IUGR is reported in 3–8% of the general population [4,5,6,7]. IUGR may be caused by fetal, placental or maternal pathology and is reported to be the second most common cause of perinatal death, accounting for 5–10% of all pregnancies [1,2,3]. In order to accurately monitor fetal growth, ultrasound biometry (measuring the circumference of the head and abdomen and the length of the femur) should be performed every two to three weeks and the dynamics should be plotted on growth curves showing the occurrence of observed changes in fetal growth [8,9]. When IUGR fetal hypoxia is detected or suspected, the results of Doppler measurements are an integral part of monitoring [10,11,12,13].

Doppler measurements are usually performed in the following arteries: the umbilical artery (UA), the middle cerebral artery (MCA), the uterine arteries (UAs) and the ductus venosus (DV). The measurements of these blood vessels provide different information [5,11,12,13,14]. The blood flow in the UA reflects the blood flow in the placenta. An increased resistance in the UA is observed in the presence of placental pathology. In terms of the MCA, the higher the resistance, the better the level of fetal oxygenation; the lower the resistance, the greater the risk of fetal hypoxia and the centralization of blood flow, and therefore the worse the fetus’s condition is likely to be. The cerebroplacental ratio (CPR), which is the ratio between the fetal middle cerebral artery pulsatility index (MCA-PI) and the umbilical artery pulsatility index (UA-PI), offers a better reflection of the condition of the fetus, rather than the assessment of a single vessel [15,16].

## 2. Case Report

We present a case of a 29-year-old woman G2 P2 with fetal IUGR. The pregnancy was uncomplicated and the fetus’s growth was normal for the gestational age. The prenatal test performed in the first trimester showed a low genetic risk. The investigation of the fetus at 34 gestational weeks was carried out at the Clinic of Obstetrics and Gynecology of Vilnius University Hospital Santaros Clinics. The biometry showed an estimated fetal weight (EFW) of 1778 g (11.8 percentile), an abdominal circumference (AC) below the 5th percentile with normal Doppler findings and normal NST. The fetal monitoring performed from 35 to 37 pregnancy weeks involved NST and Doppler examination of the fetal arteries (tibial, umbilical and middle cerebral), the ductus venosus and uterine arteries (Figure 1). NST and Doppler parameters for all the observed arteries were normal until 36 weeks and 5 days. The first pathological change was registered at this time, the tibial artery PI registering >95th percentile (Figure 2). At 37 weeks and 0 days, a pathological decrease of MCA-PI <5th percentile was registered. UA-PI remained normal. The tibial artery PI remained >95th percentile (Figure 3). At 37 weeks and 2 days, a male newborn was delivered, weighing 2070 g (<3 percentile), with a height of 48 cm, 9/9 Apgar points and the amniotic fluid clear with a pH of 7.4. In conclusion, this was a case of fetal growth restriction monitored by supplementary evaluation of the peripheral artery (the tibial artery) Doppler, which appeared to be the first indicator of the deteriorating condition of the fetus.

## 3. Discussion

The normal fetal development in the uterus depends on placental function. During pregnancy, metabolism between the mother and the fetus occurs through the placenta. At week 11 of pregnancy, the placenta attaches to the uterine wall, then, at about 12–13 weeks, increased blood supply enters the fetus through the spiral arteries, the supply increase continuing up to 24 weeks of pregnancy, when it reaches 993 mL/min/kg. Then, at 34 weeks, it drops to 360 mL/min/kg, and at 38 weeks, when the fetus is term, the blood supply is reduced threefold, to 296 mL/min/kg. Therefore, in late pregnancy the fetus can tolerate the hypoxic state for a shorter period of time than preterm fetuses [17].

The cardiovascular system of the fetus with chronic hypoxia adapts and gives priority to blood supply to the brain, myocardium and kidneys [18]. The fetal endocrine, metabolic and cardiovascular systems try to adapt, but the fetal demand for nutrients is not fully met, resulting in a high incidence of intrauterine deaths, especially in early IUGR [2,3,4].

Taking into consideration the fact that there is no specific treatment for IUGR, a continuous monitoring of the fetal IUGR condition is crucial for the timely detection of changes in fetal development. If the fetus is found to be in poor condition, the pregnancy should be terminated [12,19,20]. In early IUGR, the circulatory changes typically progress to the venous Doppler changes, whereas in late IUGR, very subtle circulatory changes and the deterioration of biophysical properties may remain unnoticed [21].

The TRUFFLE study examined IUGR over a period of up to 32 weeks of gestation, monitored DV and cardiotocography (CTG), and found that the optimal time for IUGR fetal birth should be determined according to CTG and early and late venous Doppler monitoring [22]. A comprehensive analysis of the two-year outcomes of the GRIT and TRUFFLE studies concluded that the computerized CTG and ductus venosus measurements by Doppler were found to be the best observations for early IUGR monitoring [18,23].

In late IUGR, ultrasound findings may show a normal blood flow in the umbilical cord which results in a disguised disease because the normal blood flow recorded in such cases does not directly reflect the fetal condition but is an indicator of placental blood flow [10,12,24,25,26,27]. Even though blood flow in the umbilical cord is found to be normal, the fetus is likely to have already developed a slight centralization of MCA while its CPR appears to be normal. The fetus may already be undergoing nutrient and oxygen deficiency, especially in the peripheral blood vessels [13,28]. Moreover, MCA can be unexpectedly disrupted four days before fetal death, on average [29]; however, it is still considered to be the most important Doppler examination parameter for monitoring the condition of the fetus [28].

Depending on the time of pregnancy, the fetus responds to hypoxia in a specific way due to the immaturity of the fetal cardiovascular and other systems [30,31].

Dawes and colleagues [32] were the first researchers to describe the centralization of fetal blood flow from the periphery to the central organs in the case of sheep fetuses suffering from hypoxia. It was explained as a compensatory mechanism—an adaptation to the changed intrauterine conditions of the fetus. The responses of sheep fetuses to hypoxia differed depending on the time of pregnancy. Peripheral vasoconstriction resulting from sympathetic leakage is usually already regulated in late pregnancy fetuses when there is resistance to the effect of vasodilators, such as nitric oxide (NO). The studies showed that the treatment of sheep fetuses with NO synthase inhibitors at base conditions during late pregnancy induced a general peripheral vasoconstriction and a marked increase in fetal arterial blood pressure. Fetal life in late pregnancy was balanced by the tone of NO expansion to properly maintain fetal arterial pressure. Cardiovascular changes, such as bradycardia, hypertension and femoral vasoconstriction, as well as the metabolic and endocrine changes conducing to fetal survival, were observed in sheep fetuses under hypoxic conditions. The amount of oxygen flow through the sheep fetus’s heart decreased, but as the peripheral blood vessels received less blood, increased blood supply was directed to the vital organs, such as the brain [29,30,31,32].

Similar changes are likely to occur in human fetuses undergoing intrauterine hypoxia. However, there are very few studies that have examined fetal peripheral blood flow under hypoxia. The examination of the fetal response to hypoxia by measuring peripheral vascular blood flow and comparing the UA showed a significant increase of the PI in the femoral artery (FA), while no changes were observed in the UA. This shows that the FA PI reflects peripheral perfusion [25,26,33]. A study of fetuses from 23 to 42 weeks of gestation [34] revealed that in the presence of fetal circulatory disorders, the measurement of tibial artery PI allows for the detection of circulatory changes. In IUGR fetuses, the elevated a. tibialis PI was detected from 36 weeks [35,36].

Fetal ultrasound biometry, used to assess changes in fetal development, is usually conducted every two weeks, while with fetal Doppler examination, monitoring can be made more frequently and changes detected more quickly. The researchers [30,37] who studied peripheral blood flow in the fetus concluded that the femoral artery blood flow test should not be used to assess intrauterine fetal status. However, it should be taken into account that the study was performed on fetuses up to 35 weeks of age, therefore the results obtained should receive critical evaluation.

## 4. Conclusions

The most common studies of the fetal intrauterine state focus on the central blood vessels rather than the peripheral ones. A response of sheep fetuses’ peripheral blood vessels to intrauterine hypoxia has been described by some authors. Elevated PIs in the human IUGR fetal tibial artery can be observed from 36 weeks of gestation.

Based on fetal circulatory changes and adaptation mechanisms in late IUGR, we hypothesize that peripheral vascular changes (in the tibial artery) may more accurately reflect the onset of deterioration in intrauterine fetal status. Such changes could provide an indication of early intrauterine hypoxia in late IUGR fetuses and perhaps even serve as an indicator of the time to completion of pregnancy, so that peripheral blood flow monitoring ought to be employed alongside other techniques already in use.

## Figures and Tables

**Figure 1 medicina-57-01036-f001:**
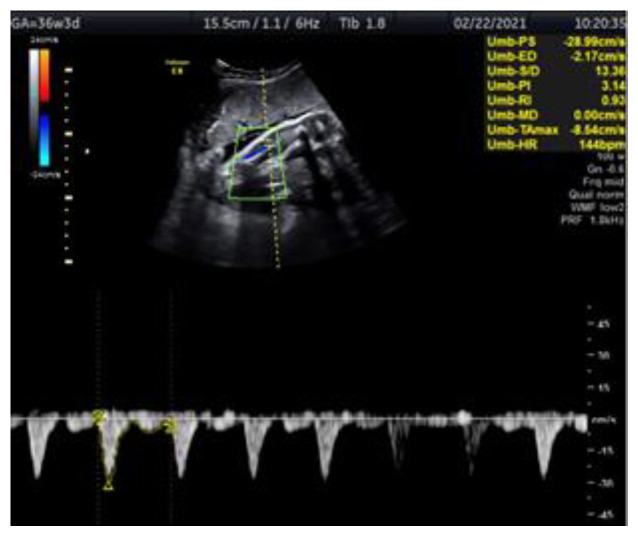
Doppler ultrasound assessment of the fetal tibial artery in IUGR fetus at 36 weeks and 3 days. The figure shows the fetal leg with the tibial artery examined by colour Doppler, normal PI (PI-3.1).

**Figure 2 medicina-57-01036-f002:**
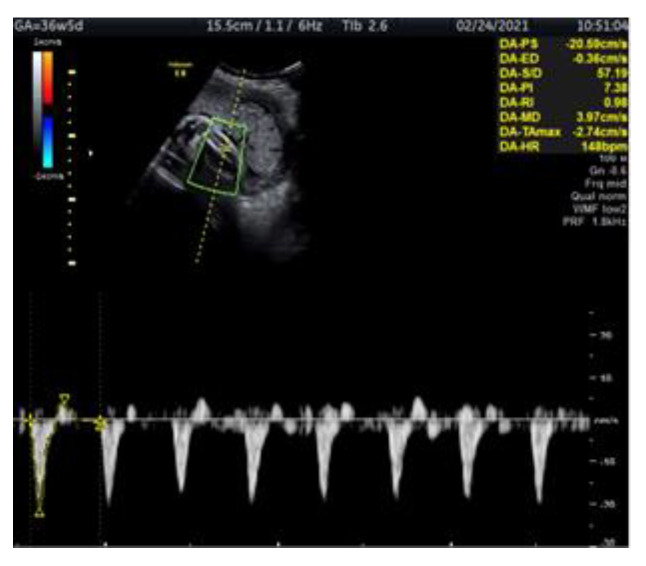
Doppler ultrasound assessment of the fetal tibial artery in IUGR fetus at 36 weeks and 5 days. The figure shows the fetal leg with the tibial artery examined by colour Doppler, PI > 95th percentile (PI-7.3).

**Figure 3 medicina-57-01036-f003:**
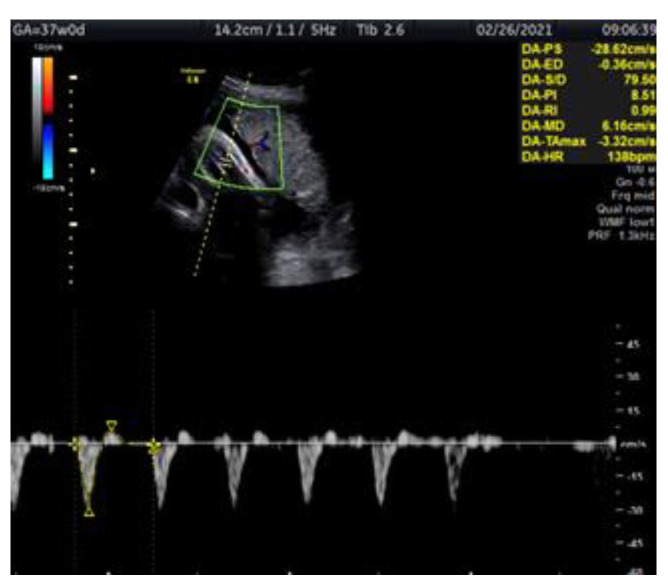
Doppler ultrasound assessment of the fetal tibial artery in IUGR fetus at 37 weeks and 0 days. The figure shows the fetal leg with the tibial artery examined by color Doppler, reverse flow, PI >95th percentile (PI-8.5).

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
