# Peer review of "Doppler Ultrasonography of the Fetal Tibial Artery in High-Risk Pregnancy and Its Value in Predicting and Monitoring Fetal Hypoxia in IUGR Fetuses"

_medicina, 2021, doi:10.3390/medicina57101036_

Round 1

Reviewer 1 Report

The intrauterine growth "retardation" - would you consider to use word "restriction" instead?

 Final English proofreading recommended.

Have no more comments.

Author Response

Dear Reviewer,   Thank you very much for your valuable observations.

CORRECTED: The intrauterine growth "retardation" - would you consider to use word "restriction" instead?

The answer to the reviewer Topic 4: it was the first indicator. "Doppler which appeared to be the first indicator of the deteriorating condition of the fetus."

Sincerely,

Kristina Norvilaite

Reviewer 2 Report

There are still issues that should be modified.

1. Abstract - To avoid repeating information, it will be suited to compress these two phrases in one “This paper describes the clinical presentation of early detection of late IUGR hypoxia. Blood flow changes in the tibial artery were the first early signs of the aggravating fetal condition.”

2. You have removed some references (from 38-55). But you have used reference 51 in here “comprehensive analysis of the two-year out-comes outcomes of the GRIT and TRUFFLE studies concluded that the computerized CTG and ductus venosus-venosus measurements by Doppler were found to be the best observations for early IUGR monitoring [18, 51].” Reference 51 is about GRIT study, mentioned in the above-cited paragraph. Please correct it.

3. Please insert references for the following paragraphs:

  1. Therefore, a consistent and continuous monitoring of the fetal development is crucial for early diagnosis. Following the guidelines, the surveillance of the middle cerebral artery, umbilical artery Doppler and the cerebroplacental ratio with cardiotocography CTG/NST in IUGR is performed.
  2. The blood flow in UA reflects the blood flow in the placenta. An increased resistance in UA is observed in the presence of placental pathology. In terms of MCA, the higher the resistance, the better fetal oxygenation, the lower the resistance, the worse the condition of the fetus, all of which indicate fetal hypoxia and the centralization of blood flow. The cerebroplacental ratio (CPR) which is the ratio between the fetal middle cerebral artery pulsatility index (MCA-PI) and umbilical artery pulsatility index (UA-PI) and better reflects the condition of the fetus rather than the assessment of a single vessel.
  3. The normal fetal development in the uterus depends on the placental function. During pregnancy, the metabolism between the mother and the fetus occurs through the placenta. At week 11 of pregnancy, placenta attaches to the uterine wall, at about 12-13 weeks the increased blood supply enters the fetus through the spiral arteries, which continues up to 24 weeks of pregnancy reaching 993 ml / min / kg, then at 34 weeks it drops to 360 ml / min / kg, and at 38 weeks when the fetus is term, the blood supply is reduced threefold, to 296 ml / min / kg. Therefore, in late pregnancy the fetus can tolerate the hypoxic state for a shorter period of time than preterm fetuses.

4. During case presentation you have said: ”NST and Doppler parameters of all the observed arteries were normal until 36 weeks 3 days when the decreased RI (Resistance Index) < 5th percentile in the middle cerebral artery (MCA) was registered. The other arteries and ductus venosus (DV) remained normal. The second pathological change was registered at 36 weeks 5 days, when the tibial artery PI (Pulsatility Index) was registered >95 percentile”. Then, in conclusion, you said ”In conclusion, this was a case of the fetal growth retardation monitored by supplementary evaluation of the peripheral artery (the tibial artery) Doppler which appeared to be one of the first indicators of the deteriorating condition of the fetus.” First or second?

Author Response

Dear Reviewer,

Thank you for you valuable observations.

  1. CORRECTED. Abstract - To avoid repeating information, it will be suited to compress these two phrases in one “This paper describes the clinical presentation of early detection of late IUGR hypoxia. Blood flow changes in the tibial artery were the first early signs of the aggravating fetal condition.”

 2. CORRECTED. You have removed some references (from 38-55). But you have used reference 51 in here “comprehensive analysis of the two-year out-comes outcomes of the GRIT and TRUFFLE studies concluded that the computerized CTG and ductus venosus-venosus measurements by Doppler were found to be the best observations for early IUGR monitoring [18, 51].” Reference 51 is about GRIT study, mentioned in the above-cited paragraph. Please correct it.

3. CORRECTED. Please insert references for the following paragraphs:

  1. CORRECTED. Therefore, a consistent and continuous monitoring of the fetal development is crucial for early diagnosis. Following the guidelines, the surveillance of the middle cerebral artery, umbilical artery Doppler and the cerebroplacental ratio with cardiotocography CTG/NST in IUGR is performed.
  2. CORRECTED. The blood flow in UA reflects the blood flow in the placenta. An increased resistance in UA is observed in the presence of placental pathology. In terms of MCA, the higher the resistance, the better fetal oxygenation, the lower the resistance, the worse the condition of the fetus, all of which indicate fetal hypoxia and the centralization of blood flow. The cerebroplacental ratio (CPR) which is the ratio between the fetal middle cerebral artery pulsatility index (MCA-PI) and umbilical artery pulsatility index (UA-PI) and better reflects the condition of the fetus rather than the assessment of a single vessel.
  3. CORRECTED. The normal fetal development in the uterus depends on the placental function. During pregnancy, the metabolism between the mother and the fetus occurs through the placenta. At week 11 of pregnancy, placenta attaches to the uterine wall, at about 12-13 weeks the increased blood supply enters the fetus through the spiral arteries, which continues up to 24 weeks of pregnancy reaching 993 ml / min / kg, then at 34 weeks it drops to 360 ml / min / kg, and at 38 weeks when the fetus is term, the blood supply is reduced threefold, to 296 ml / min / kg. Therefore, in late pregnancy the fetus can tolerate the hypoxic state for a shorter period of time than preterm fetuses.

4. During case presentation you have said: ”NST and Doppler parameters of all the observed arteries were normal until 36 weeks 3 days when the decreased RI (Resistance Index) < 5th percentile in the middle cerebral artery (MCA) was registered. The other arteries and ductus venosus (DV) remained normal. The second pathological change was registered at 36 weeks 5 days, when the tibial artery PI (Pulsatility Index) was registered >95 percentile”. Then, in conclusion, you said ”In conclusion, this was a case of the fetal growth retardation monitored by supplementary evaluation of the peripheral artery (the tibial artery) Doppler which appeared to be one of the first indicators of the deteriorating condition of the fetus.” First or second?

The last item of your response will be clarified ASAP.

Sincerely,

Kristina Norvilaite